# Attributes of *Lactobacillus acidophilus* as Effected by Carao (*Cassia grandis*) Pulp Powder

Jhunior Marcia [1], Ricardo Santos Aleman [2], Ismael Montero-Fernández [3,*], Daniel Martín-Vertedor [4], Víctor Manrique-Fernández [5], Marvin Moncada [6] and Aryana Kayanush [2]

1. Faculty of Technological Sciences, Universidad Nacional de Agricultura Road to Dulce Nombre de Culmí, Km 215, Barrio El Espino, Catacamas 16201, Honduras
2. School of Nutrition and Food Sciences, Louisiana State University Agricultural Center, Baton Rouge, LA 70803, USA
3. Department of Chemical Engineering and Physical Chemistry, Area of Chemical Engineering, Faculty of Sciences, University of Extremadura, Avda. de Elvas, s/n, 06006 Badajoz, Spain
4. Technological Institute of Food and Agriculture CICYTEX-INTAEX, Junta of Extremadura, Avda. Adolfo Suárez s/n, 06007 Badajoz, Spain
5. Área de Nutrición y Bromatología, Departamento de Producción Animal y Ciencia de los Alimentos, Escuela de Ingenierías Agrarias, Universidad de Extremadura, Avda. Adolfo Suárez s/n, 06007 Badajoz, Spain
6. Department of Food, Bioprocessing & Nutrition Sciences and the Plants for Human Health Institute, North Carolina Research Campus, North Carolina State University, Kannapolis, NC 27599, USA
* Correspondence: ismonterof@unex.es

**Abstract:** This study aimed to examine the prebiotic effect of *Carao* (*Cassia grandis*) pulp powder on the probiotic characteristics of *Lactobacillus acidophilus* regarding the viability, enzymatic activity, lysozyme resistance, bile and acid tolerances, and tolerance to gastric juices. *Carao* powder was used at 0% (control), 1%, 2%, and 3% (*w/v*). Acid and lysozyme tolerance were determined at 0, 30, 60, 90, and 120 min of incubation, whereas bile tolerance was analyzed at 0, 4, and 8 h. The gastric juice tolerance was determined at pH 2, 3, 4, 5, and 7 during 0 and 30 min of incubation. The protease was evaluated at 0, 12, and 24 h of incubation. The bacterial viability experiment was carried out for 10 h, taking readings every hour. Low-acidity conditions were used, and no significant differences were found between the control and the different *Carao* concentrations added to the *L. acidophilus* viability study. The *Carao* samples at 2% and 3% had significantly ($p < 0.05$) higher counts for bile and lysozyme resistance and higher protease activity when compared to control samples. On the other hand, *Carao* addition did not impact bacterial viability, acid tolerance, and gastric juice resistance. Thus, *Carao* pulp powder at different concentrations could act as a prebiotic source to enhance the development of *L. acidophilus* during gastrointestinal digestion.

**Keywords:** *Lactobacillus acidophilus*; *Carao*; enzymatic activity; lysozyme; gastric juices

## 1. Introduction

In recent decades, considerable scientific evidence has shown that the interaction of microbiota with gastrointestinal digestion in humans is fundamental for health balance. In this sense, studies that have been conducted that assess the use of microorganisms such as *Lactobacillus acidophilus* have greatly benefited human health [1]. Clinical trials have demonstrated that this microorganism could prevent and treat diarrhea (infantile, acute, and associated with antibiotics) in children [2]. *Lactobacillus acidophilus* is also effective in treating symptoms accompanying lactose intolerance, inflammatory bowel diseases, modulation of the immune system, and colon cancer [3]. Furthermore, many researchers have carried out studies for the inhibition of different digestive cancers when these microorganisms are administered in adequate concentration. Isazadeh et al. (2021) [4] indicated that *L. acidophilus* could inhibit the viability of colorectal cancer of Caco-2 cell line, increasing the survival rate of the patients.

Certain foods, due to their chemical composition, are implemented as a prebiotic source, because these are considered essential for people's health. Therefore, prebiotic foods play an important role in the microbiota, and are beneficial for the human gastrointestinal tract [5]. Prebiotics are defined as non-digestible compounds that serve to modulate the composition and activity of the intestinal microbiota, which are metabolized by microorganisms in the intestine, and confer a beneficial effect on the host [6]. Prebiotics are found in many fruits and vegetables, especially those that contain complex carbohydrates, such as fiber and resistant starch [6], or non-digestible compounds, such as inulin, that can stimulate the multiplication of microorganisms in the colon, modulating the intestinal microbiota. Different investigations with prebiotic foods such as honey [7], meat products [8], fermented soybean [9], or vegetable milk [10] have been shown to enhance microbial development. As a result, new prebiotic sources are demanded by consumers to be incorporated into their daily diet, and food industries are prioritizing the development of these prebiotic products. Furthermore, prebiotics can stimulate the growth of healthy bacteria in the intestine, reducing the risk of developing certain diseases [11].

*Cassia grandis* (*Carao*) fruits have been used in alternative medicine due to their characteristic effect in humans and chemical composition [12]. These fruits have shown considerable amounts of alkaloids, flavonoids, and phenols, and excellent antioxidant capacity [13]. In the leaves, *Carao* has shown substantial amounts of phenolic compounds such as Grandisina, Kaempferol, Quercetin, and Flavonol [14]. Besides, the phytochemical characteristics of this fruit, it has great antidiabetic potential due to its trypsin inhibitory effect [15]. In addition, *Carao* has been shown to improve acid and bile tolerance of *Streptococcus thermophilus* and *Lactobacillus bulgaricus* [16]. Due to all these characteristics, this fruit could be used as a potential prebiotic against certain microorganisms. As a result, the current study aims to study the prebiotic effect of *Carao* powder on enzymatic activity, lysozyme resistance, bile and acid tolerances, and tolerance to gastric juices of *Lactobacillus acidophilus*, to determine its potential to promote resistance in the digestive system from the mouth to the intestines.

## 2. Materials and Methods

### 2.1. Plant Material

The *C. grandis* (*Carao*) fruit was gathered from the Guapinol Biological Reserve, Marcovia Municipality, Choluteca Department (Honduras), between August and September 2021. The pulp was separated, and a solution of *Carao* pulp (10% $w/w$) was prepared and then kept cryogenically ($-80$ °C). The obtained *Carao* aqueous solution was then lyophilized (LIOTOP model L 101) for 48 h at a temperature of $-75$ °C and a chamber pressure of 0.1 to 0.5 Pa. The freeze-dried *Carao* pulp powder was kept in plastic bags for further use.

### 2.2. Experimental Design

The viability; acid, bile, lysozyme, and gastric juice tolerances; and protease activity of *Lactobacillus acidophilus* LYO 50 (Danisco, Dairy Connection, Madison, WI, USA) as affected by *Carao* powder were examined at 0% (control), 1%, 2%, and 3%. The bacterial viability was studied in MRS broth. Acid tolerance was determined by adjusting the pH to 2, whereas bile tolerance was examined with Oxgall 0.3% ($w/v$) in MRS broth. Lysozyme resistance was investigated in an electrolyte solution with lysozyme (100 mg/L), while gastric juice tolerance was analyzed using pepsin and NaCl. Protease activity was determined spectrophotometrically at 340 nm in skim milk with o-phthaldialdehyde reagent. The microbial growth was determined at 0, 2, 4, 6, 8, and 10 h of incubation. Acid tolerance was determined at 0, 5, and 15 min, whereas bile tolerance was analyzed at 0, 4, and 8 h of incubation. Lysozyme tolerance was determined at 0, 1, and 2 h of incubation, while gastric juice tolerance was determined at pH 2, 3, 4, 5, and 7. The protease activity was evaluated at 0, 12, and 24 h of incubation. L. acidophilus was incubated anaerobically (37 °C). The log counts were measured in MRS agar with duplicate readings. All experiments were carried out in triplicate.

### 2.3. Analytical Method

#### 2.3.1. Bacterial Viability

The viability of *L. acidophilus* was examined by the procedure suggested by Lin and Young (2000) [17], with some changes. The culture (inoculation of 10% (*v/v*)) was inoculated in MRS broth (CriterionTM, Hardy Diagnostics, Santa Maria, CA, USA) containing 0.5% lactose with 0.2% (*w/v*) sodium thioglycolate (Sigma-Aldrich, St. Louis, MO, USA) and the pH was adjusted to 6.5. The cultured broths were incubated at 37 °C. An 11 mL sample was collected at several periods (0, 1, 2, 3, 4, 5, 6, 7, 8, 9, and 10 h), 10-fold diluted in peptone water, and plated in duplicate. Bacterial viability was evaluated in presence and absence of *Carao* pulp at three different concentrations.

#### 2.3.2. Bile Tolerance

The bile tolerance of *L. acidophilus* was evaluated using the Pereira & Gibson (2002) [18] method, with slight modification. After the culture media broth was prepared, *Carao* pulp powder, water, and 1.5 g bile salt (Oxgall salt 0.3%) were added and autoclaved. The culture (inoculation of 10% (*v/v*)) was inoculated in MRS broth (CriterionTM, Hardy Diagnostics, Santa Maria, CA, USA) containing 0.5% lactose with 0.2% (*w/v*) sodium thioglycolate (Sigma-Aldrich) and bile salt Oxgall (bovine bile) (US Biological, Swampscott, MA, USA). The cultured broths were incubated at 37 °C. An 11 mL sample was collected at several periods (0, 4, and 8 h), 10-fold diluted in peptone water, and plated in duplicate. Tolerance to specific acidity and resistance to gastric juices were evaluated in the presence and absence of *Carao* pulp at three different concentrations.

#### 2.3.3. Acid Tolerance

The acid tolerance of *L. acidophilus* was evaluated by inoculation of culture (10% (*v/v*)) into the acidified MRS broth containing 0.5% lactose, with 1 N HCl added to produce pH 2.0. This acidified MRS broth containing the culture was incubated at a temperature of 37 °C. A 1 mL sample was collected at several periods (0, 30, and 60 min). The protease activity was evaluated in the presence and absence of *Carao* pulp at three different concentrations.

#### 2.3.4. Protease Activity for Probiotics

*L. acidophilus* protease activity was evaluated by inoculation of culture (10% (*v/v*)) using the o-phthaldialdehyde (OPA) spectrophotometric test established by Oberg et al. (1991) [19]. After incubation of *L. acidophilus* in sterile skim milk [20], *L. acidophilus* was grown at 37 °C for 0, 12, and 24 h, then 2.5 mL of each sample was combined with 1 mL of distilled water and 10 mL of 0.75 N trichloroacetic acid (TCA) to give a final concentration of 7.7%. All samples were filtered using a Whatman Number 2 filter paper for 10 min at ambient conditions. A double portion of each TCA filtrate was examined by the OPA spectrophotometric test utilizing a spectrophotometer at 340 nm (Nicolet Evolution 100, Thermo Scientific; Madison, WI, USA).

#### 2.3.5. Tolerance to Simulated Gastric Juice

The tolerance of *L. acidophilus* to functional substances in synthetic gastric juice (SGJ) was tested using the method described by García-Ruiz et al. (2014), Aleman et al. (2023), and Liao et al. (2019) [21–23]. The SGJ was prepared using $H_2O$, pepsin 0.32% (Sigma-Aldrich, St. Louis, MO, USA), NaCl 0.2%, NaOH, and HCl for pH adjustment [24]. Lysozyme resistance was evaluated in the presence and absence of *Carao* pulp at three different concentrations. With 1 M HCl and 1 M NaOH, the simulated gastric juice was modified to five concentration gradients (pH 7, 5, 4, 3, and 2). The culture was inoculated (10% (*w/v*)) into SGJ, and incubated for 30 min under anaerobic conditions at 37 °C. Plates were counted at 0 and 30 min of incubation to determine live bacteria. Bacterial viability was measured by inoculating the bacteria in MRS broth, and numeration of *L. acidophilus* was determined by plating the bacteria with MRS agar. Paz et al. (2022) [16] proposed these methods as well.

2.3.6. Lysozyme Tolerance

The *L. acidophilus* resistance to lysozyme was evaluated according to Zago et al. (2011) [25], with slight modification. The electrolyte solution was used to control the lysozyme tolerance test and to imitate in vivo dispersion by saliva. Bacteria cultures were inoculated (10% (*w/v*)) into sterile electrolyte solution (SES) of 0.22 g·L$^{-1}$ CaCl$_2$, 6.2 g·L$^{-1}$ NaCl, 2.2 g·L$^{-1}$ KCl, and 1.2 g·L$^{-1}$ NaHCO$_3$ in the presence of lysozyme (100 mg·L$^{-1}$) (Sigma-Aldrich, CA USA). Tests comprised microbial cultures in SES without lysozyme. Bacterial counting was performed on MRS agar after incubation (72 h at 37 °C). The survival expectancy was determined by comparing the CFU·mL$^{-1}$ at 0, 30, 60, 90, and 120 min.

2.3.7. Enumeration of *L. acidophilus*

Preparation of the MRS broth of *L. acidophilus* included 1 L of distilled water being added to 55 g of MRS broth powder (Difco, Becton, Dickinson and Co., Sparks, MD, USA). Next, 1 N HCl was utilized to reduce the pH to 5.2. To thoroughly disperse the particles, this medium was boiled under stirring, as well as sterilized at 121 °C for 15 min [26,27]. Following plating into the inoculated medium, MRS broths were pipetted to various formulations using 99 mL of sterilizing phosphate buffer 0.1% (*w/v*). Following 72 h, these *L. acidophilus* plates were heated anaerobically at 37 °C. A Quebec Darkfield Colony Counter was used to calculate (Leica Inc., Buffalo, NY, USA) [16].

*2.4. Statistical Analysis*

Data were analyzed using the General Linear Model (PROC GLM) of the Statistical Analysis Systems (SAS). Differences of least square means were used to determine significant differences at *p* < 0.05 for the main effect (*Carao* pulp concentration vs. control). Data are presented as mean ± standa0072d error of means. Significant differences were determined at α = 0.05.

**3. Results and Discussion**

*3.1. Bacterial Viability*

The bacterial viability of *L. acidophilus* over 10 h of incubation after the addition of *Carao* pulp powder is shown in Figure 1. The *Carao* concentration effect and the interaction effect (*Carao* concentration × hour) were not significant (*p* > 0.05), whereas the hour effect was significant (*p* < 0.05) (Table 1). The interaction effect was not significant (*p* > 0.05), meaning that the control and *Carao* samples followed the same trend. Control and *Carao* samples increased in log counts over time. The log count increased from 8.83 to 9.38 from 0 to 10 h for control samples. All *Carao* treatments followed a comparable viability tendency to the control samples (Table 2). This is a good result, since the high antioxidant capacity of this fruit [12] at this concentration does not cause a marked inhibition of the development of *L. acidophilus*. Muramalla and Aryana (2011) [28] examined the viability of *L. acidophilus* in MRS broth, and concluded that an increase in viability was reached in the first 3 h of incubation. Paz et al. (2022) [16] also reported that *Carao* did not adversely impact the viability of *Streptococcus thermophilus* and *Lactobacillus bulgaricus*.

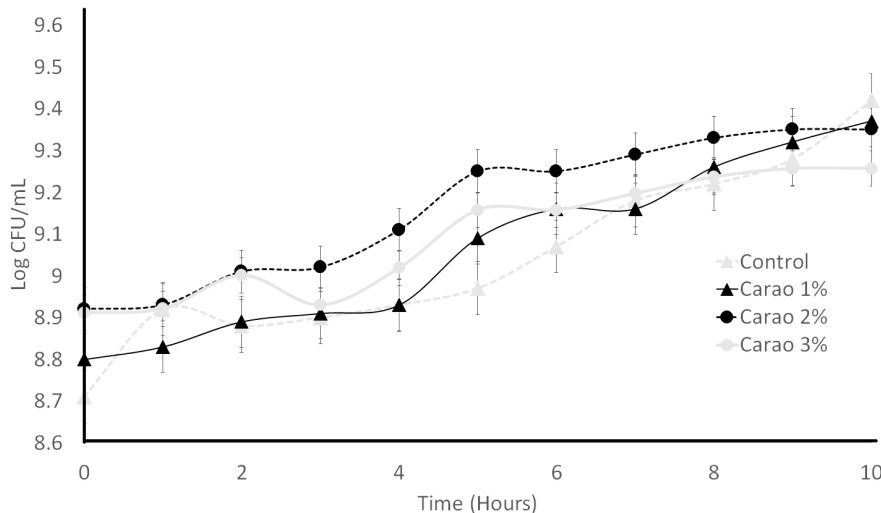

**Figure 1.** Viability of *L. acidophilus* as influenced by *Carao* concentration over 10 h.

**Table 1.** The *p*-value or F-value of *Carao* concentration, time, pH, and their interactions for bacterial viability, bile tolerance, acid tolerance, resistance to gastric juices, protease activity, and lysozyme resistance of *Lactobacillus acidophilus* LA-K.

| Effect | *L. acidophilus* LA-K |
|---|---|
| Viability | |
| *Carao* concentration | 0.0770 |
| Time (Hours) | <0.0001 |
| *Carao* concentration × time | 0.4756 |
| Bile tolerance | |
| *Carao* concentration | 0.0057 |
| Time (Hours) | <0.0001 |
| *Carao* concentration × time | 0.0045 |
| Acid Tolerance | |
| *Carao* concentration | 0.095 |
| Time (Minutes) | <0.0001 |
| *Carao* concentration × time | 0.5867 |
| Resistance to gastric juices | |
| *Carao* concentration | 0.0786 |
| pH | <0.0001 |
| *Carao* concentration × pH | 0.7845 |
| Protease activity | |
| *Carao* concentration | 0.0155 |
| Time (Hours) | <0.0001 |
| *Carao* concentration × time | 0.4021 |
| Lysozyme resistance | |
| *Carao* concentration | 0.0085 |
| Time (Minutes) | <0.0001 |
| *Carao* concentration × time | 0.3945 |

The concentration and the type of plant influence microbial growth. For the most part, medical plants' phenolic content has reported inhibitory effects due to different mechanisms of action, including the inhibition of acid production and the glucosyltransferase enzyme [29]. Plants such as *Plantago major L*, *Erythroxylum novogranatense*, *Plowman var truxillensey*, and *Camellia* (extracted using 70% ethanol) have been shown to have antimicrobial activity towards *L. acidophilus* when the extracts are diluted to 25 $\mu$g·mL$^{-1}$ and 50 $\mu$g·mL$^{-1}$ [29]. Medical plant *Tagetes eliptica* (extracted using 70% ethanol) extract has been shown to have an inhibition effect when extracts are diluted no lower than

62.5 mg·mL$^{-1}$ [30]. Similarly, at concentrations greater than 1%, clove extract was reported to have antimicrobial activity toward *L. acidophilus* in MRS broth [31].

**Table 2.** Least squares means for bacterial viability, bile tolerance, acid tolerance, resistance to gastric juices, protease activity, and lysozyme resistance of *Lactobacillus acidophilus* LA-K as influenced by *Carao* concentration.

| Test | *L. acidophilus* **LA-K** |
|---|---|
| Bacterial Viabiliy | |
| *Carao* 0% (Control) | NS |
| *Carao* 1% | NS |
| *Carao* 2% | NS |
| *Carao* 3% | NS |
| Bile tolerance | |
| *Carao* 0% (Control) | 10.20 [A] |
| *Carao* 1% | 10.17 [A] |
| *Carao* 2% | 10.24 [B] |
| *Carao* 3% | 10.33 [C] |
| Acid Tolerance | |
| *Carao* 0% (Control) | NS |
| *Carao* 1% | NS |
| *Carao* 2% | NS |
| *Carao* 3% | NS |
| Resistance to gastric juices | |
| *Carao* 0% (Control) | NS |
| *Carao* 1% | NS |
| *Carao* 2% | NS |
| *Carao* 3% | NS |
| Protease activity | |
| *Carao* 0% (Control) | 0.300 [A] |
| *Carao* 1% | 0.321 [A] |
| *Carao* 2% | 0.339 [B] |
| *Carao* 3% | 0.355 [B] |
| Lysozyme resistance | |
| *Carao* 0% (Control) | 6.06 [A] |
| *Carao* 1% | 6.15 [A] |
| *Carao* 2% | 6.36 [B] |
| *Carao* 3% | 6.68 [B] |

A, B: Means within the same column along with the same test with different letters differ statistically ($p < 0.05$).

### 3.2. Bile Tolerance

The effect of *Carao* pulp power on *L. acidophilus'* tolerance to bile (to resist Oxgall salt) in MRS broth is shown in Figure 2. The main effects (*Carao* concentration and time) and interaction effect were significant ($p > 0.05$) (Table 1). The interaction effect was not significant ($p > 0.05$), meaning that the control and *Carao* samples did not follow the same trend (Table 2). Control and 1% and 2% *Carao* samples decreased in log counts over time, whereas 3% *Carao* samples increased in log counts over time. Theegala et al. (2021) [32] reported a similar growth for *L. acidophilus* evaluated in MRS broth. Bile salts are known to alter eukaryotic gene expression, denature proteins, damage membranes, chelate calcium and iron, and disrupt DNA in probiotics' immunity activity [33]. At 5, 6, 7, and 8 h, 2% and 3% *Carao* broths had significantly ($p > 0.05$) higher counts than control samples. Adding inulin to yogurt enhanced the capacity of *L. acidophilus* in yogurt to resist bile salts [34], and incorporating flaxseed into MRS broth with 0.3% Oxgall salt also improved the survivability of *L. acidophilus* [32]. The addition of 5.3 g·L$^{-1}$ of *Carao* improved resistance to Oxgall salt (0.03%) in M17 and MRS broth for *Streptococcus thermophilus* and *Lactobacillus delbrueckii* ssp. *bulgaricus*, respectively [16]. Alginate–milk microspheres can encapsulate *L. bulgaricus* and increase the survivability for 1 and 2 h in 1% and 2% porcine bile salt solutions [35]. In *Cassia fistula*, polysaccharides with encapsulating properties have been reported [36]. It is

possible that polysaccharides in *Carao* could have an encapsulating effect on *L. acidophilus*, resulting in improved bile tolerance.

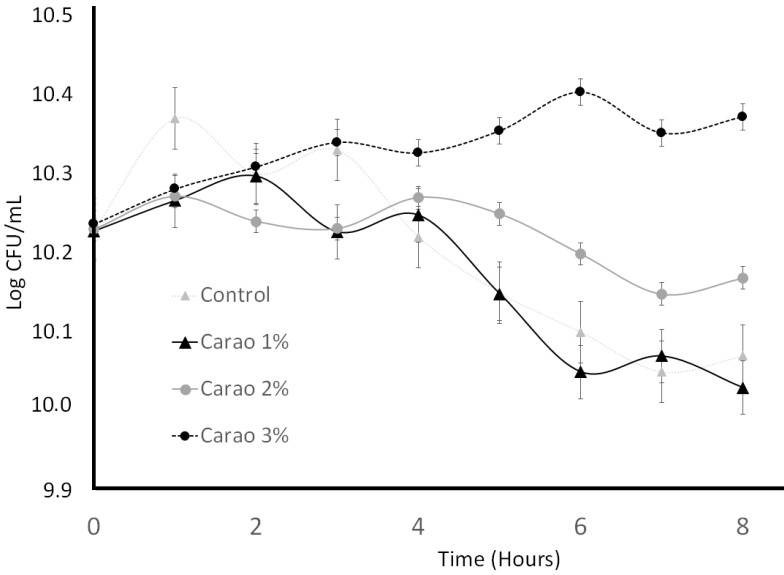

**Figure 2.** Bile tolerance of *L. acidophilus* as influenced by *Carao* concentration over 8 h.

### 3.3. Acid Tolerance and Resistance to Gastric Juices

The acid tolerance and resistance of *L. acidophilus* to gastric juices are shown in Figure 3 and Figure 4, respectively. The results at different pH values are shown in Figure 4. Acid tolerance was examined through a 15 min bacterial count to examine the effects of *Carao* concentration on the survival of *L. acidophilus* under stomach acid condition. Under normal conditions, the transit time through the gastrointestinal system ranges from 2 to 4 h, and varies depending on the individual [37]. The gastric juice resistance was analyzed at different pH values (2, 3, 4, 5 and 7) for bacterial viability, to investigate the influence of *Carao* concentration on the survival of *L. acidophilus* in the different parts of the digestive system (esophagus, small intestine, and large intestine). For acid tolerance, the *Carao* concentration and interaction effect (*Carao* concentration × time) were not significant ($p < 0.05$), whereas the time effect was significant ($p > 0.05$) (Table 1). The interaction effect was not significant ($p > 0.05$), meaning that the control and *Carao* samples followed the same trend (Table 2). Control and *Carao* samples decreased in log counts over time. The log counts decreased from 0 to 10 min, and remained stable from 10 to 15 min for the *Carao* treatments and control. For gastric juice resistance, the *Carao* concentration and interaction effect (*Carao* concentration and pH) were not significant ($p < 0.05$), whereas the pH effect was significant ($p > 0.05$) (Table 1). The log counts were lower from pH 2 to 4 and higher from pH 5 to 7 for the *Carao* treatments and control. Not surprisingly, these results are consistent with other studies showing that *Lactobacillus* strains showed lower counts when exposed to pH values of 4.0, and higher viability at higher pH values [38]. It was observed that adding *Carao* into MRS broth did not influence the acid tolerance and gastric juice resistance of *L. acidophilus*. The high levels of H⁺ ions can disrupt hydrogen bonding, promoting cell denaturation and destroying activity by producing complications for associated cell membrane energetics [39]. Probiotics must persist under the acidic gastric conditions from the digestive system to colonize in the small intestine. Thereby, functional ingredients must, at least, not negatively affect the probiotics in order to survive the acidic gastric environment. Paz et al. (2022) [16] reported that *C. grandis* improved the acid tolerance of *S. thermophilus*, and it did not impact the acid tolerance of *L. bulgaricus*.

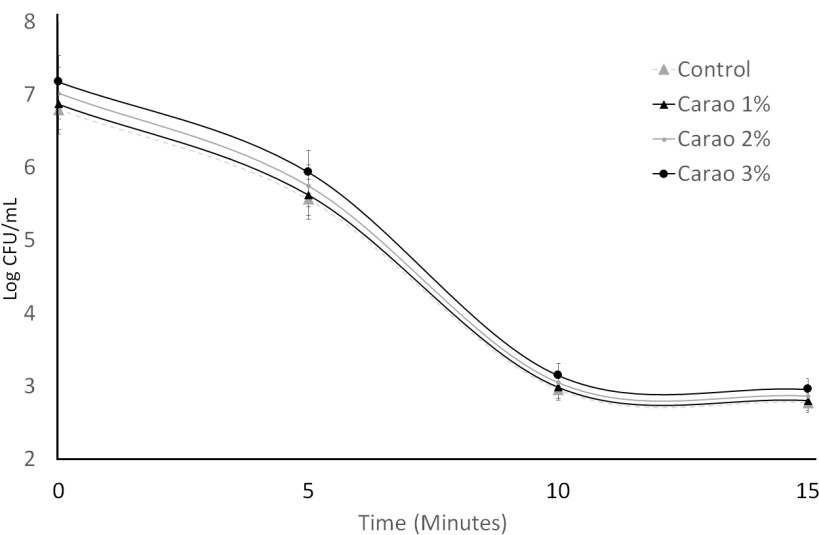

**Figure 3.** Acid tolerance of *L. acidophilus* as influenced by *Carao* concentration over 15 min. Average of three replicates. Error bars represent SE.

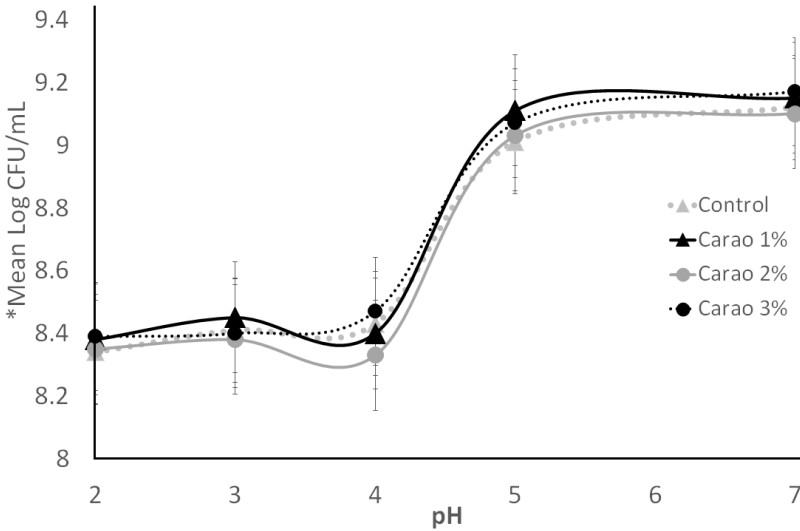

**Figure 4.** Gastric juices resistance of *L. acidophilus* as influenced by *Carao* concentration over different pH levels (2,3,4,5, and 7). * Average mean log CFU/mL of 0 min and 30 min. Average of three replicates. Error bars represent SE.

### 3.4. Protease Activity

Proteolysis in fermented milk is of great importance for several aspects: it can determine the survival of the probiotic cultures, it contributes to the formation of flavor and odor compounds, it confers rheological properties, and it allows the formation of bioactive peptides [40]. The protease activity of *L. acidophilus* is shown in Figure 5. Proteolysis is the degradation of proteins by the action of the proteolytic system of lactic acid bacteria (LAB), which produces small peptides and free amino acids that are essential for probiotic growth and activity [41]. The *Carao* concentration and time effects were significant ($p > 0.05$), whereas the interaction effect (*Carao* concentration × time) was not significant ($p < 0.05$) (Table 1). The interaction effect was not significant ($p > 0.05$), meaning that the control and *Carao* samples followed the same trend. Control and *Carao* samples increased in log counts over time. The protease activity of *L. acidophilus* showed an increase after 24 h for control and *Carao* treatments (Table 2). The proteolytic activity of *L. acidophilus* is mainly because of the synthesis of serine-like proteinase [42]. *Carao* treatments showed no significant difference ($p > 0.05$) from control samples at 0 h and 12 h, whereas 2% *Carao* and 3% *Carao* had

significantly higher protease activity at 24 h. Paz et al. (2022) [16] also reported that *Carao* pulp increases the protease activity of *Streptococcus thermophilus* and *Lactobacillus bulgaricus* after 24 h in skim milk. During milk fermentation, the proteolytic system of probiotic cultures plays a key role [40], since the proteolysis carried out by each microorganism is initiated by a single extracellular proteinase. Danisco, Dairy Connection, Madison, WI was studied in absence (control) and in presence of three different concentrations of *Carao* pulp powder (1%, 2%, and 3%). Acid and lysozyme tolerance were determined.

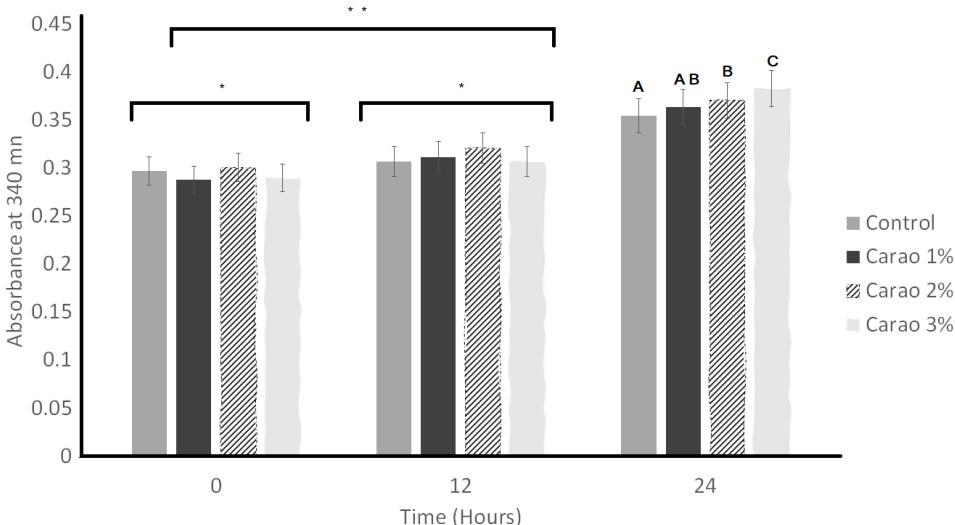

**Figure 5.** Protease activity of *L. acidophilus* as influenced by *Carao* concentration over 24 h. * Average of three replicates. [A–C] Values with different letters are significantly different between control and *Carao* treatments ($p < 0.05$). Error bars represent SE. * No significant differences between control and *Carao* treatments ($p < 0.05$). ** No significant differences between 0 h and 12 h ($p < 0.05$).

### 3.5. Lysozyme Resistance

Resistance to lysozyme is shown in Figure 6. Lysozyme is a crucial element of antimicrobial activity in saliva, making it an essential component of mouth immune activity. The mechanism of action of this enzyme on, especially, Gram-positive bacteria is by hydrolyzing 1,4-beta-linkages between N-acetylglucosamine and N-acetylmuramic acid in the bacterial membrane [43]. The *Carao* concentration and time effects were significant ($p > 0.05$), whereas the interaction effect (*Carao* concentration × time) was not significant ($p < 0.05$) (Table 1). The interaction effect was not significant ($p > 0.05$), meaning that the control and *Carao* samples followed the same trend. Control and *Carao* samples decreased in log counts over time. For control and *Carao* treatments, the log counts decreased from 0 to 60 min, and remained stable from 60 to 120 min. The 2% and 3% *Carao* electrolyte dispersions reported significantly ($p > 0.05$) higher viability than control samples (Table 2). *Carao* has substantial portions of sucrose [44], and this substrate could be used for the survivability of this probiotic. Furthermore, *Carao* can also inhibit the digestive enzyme activity of pancreatic lipase [15]. As a hypothesis, *Carao* could act as a barrier between the cell membrane and the lysozyme to protect *L. acidophilus*, leading to higher viability in electrolyte solutions with *Carao* treatments, and possibly inhibiting the hydrolysis of 1,4-beta-linkages between N-acetylglucosamine and N-acetylmuramic acid. Nevertheless, this mechanism of action still needs to be confirmed, and more research on this topic is encouraged.

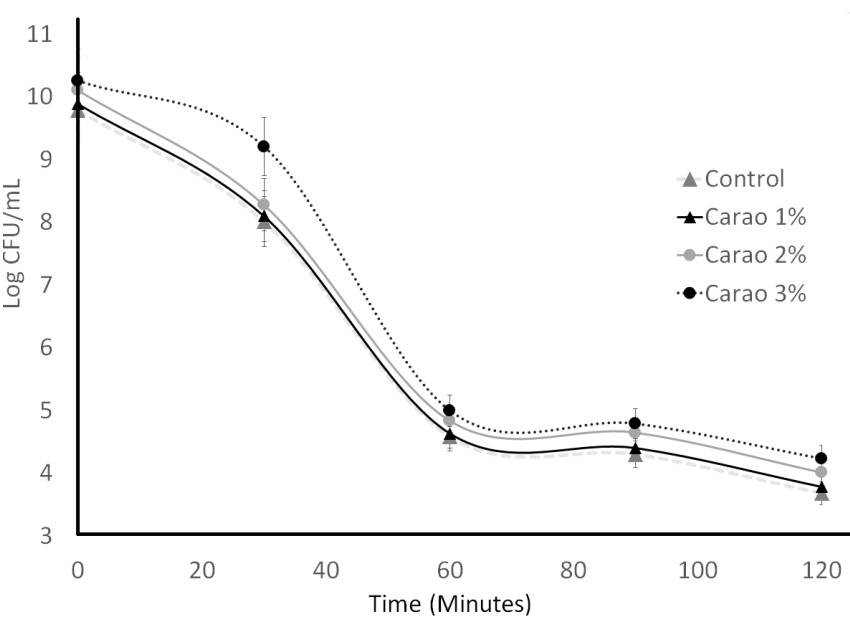

**Figure 6.** Lysozyme resistance of *L. acidophilus* as influenced by *Carao* concentration over 120 min.

### 4. Conclusions

The results showed that the *Carao* fruit could be used as a potential prebiotic for *Lactobacillus acidophilus* because it does not affect bacterial viability, acid tolerance, and resistance to gastric juices, since, due to its high antioxidant capacity, there is a viability trend comparable to that of the fruit. In addition, the addition of 2% and 3% *Carao* improved bile tolerance. For the tolerance to gastric juices, there was no influence of *Carao* on the tolerance to acid or on the resistance to gastric juices of *Lactobacillus*. Regarding its protease activity, *Carao* in concentrations of 2 and 3% presented significant activity at 24 h, as well as resistance to *L. acidophilus* lysozyme in MRS broth and electrolyte solution. Due to this, *Carao* could act as a barrier between the cell membrane and the protective lysozyme of *L. acidophilus*, so *Carao* could have prebiotic properties against *L. acidophilus*. For future research, it is suggested to examine the probiotic characteristics of *Carao* in vivo, to enable its precise application in prebiotic or symbiotic scenarios.

**Author Contributions:** Conceptualization, R.S.A., J.M. and I.M.-F.; methodology, R.S.A., J.M., M.M., V.M.-F., A.K. and I.M.-F.; software, R.S.A. and I.M.-F.; formal analysis, J.M. (most of the research), R.S.A. and I.M.-F.; resources, R.S.A. and I.M.-F.; data curation, I.M.-F., R.S.A. and J.M.; writing—original draft preparation, R.S.A., J.M., V.M.-F. and I.M.-F.; writing—review and editing, R.S.A., J.M., M.M., A.K., V.M.-F., D.M.-V. and I.M.-F.; project administration, R.S.A., I.M.-F. and D.M.-V.; funding acquisition, R.S.A., A.K., D.M.-V. and I.M.-F. All authors have read and agreed to the published version of the manuscript.

**Funding:** This research was funded by the the European Regional Development Fund (FEDER) and Universidad Nacional de Agricultura (Honduras).

**Data Availability Statement:** The authors confirm that the data supporting the findings of this study are available within the article and the raw data that support the findings are available from the corresponding author, upon reasonable request.

**Acknowledgments:** We wish to thank the School of Food Sciences, Louisiana State University Agricultural Center; the Faculty of Technological Sciences, Universidad Nacional de Agricultura Road to Dulce, Catacamas, Olancho, Honduras; the Research Groups of the Junta de Extremadura (ref. GR21121); and the European Regional Development Fund (FEDER) for their help in the development of this work. This research was also funded by USDA Hatch funds (LAB94511).

**Conflicts of Interest:** The authors declare no conflict of interest.

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
