# Peer review of "Attributes of Lactobacillus acidophilus as Effected by Carao (Cassia grandis) Pulp Powder"

_fermentation, doi:10.3390/fermentation9050408_

Round 1

Reviewer 1 Report

The topic article is very interesting, and the experimental plan is well designed since a broad characterization was performed. The weakness of this work is the clear less in the exposition of the experimental design and materials and methods part, that doesn’t allow an immediate understanding of the study; furthermore conclusions are very poor . I suggest the attached revisions before publication.

Author Response

Thanks for your appreciation. We have reviewed the manuscript point by point according to the reviewer.

Reviewer 1

Comments to Author

The topic article is very interesting, and the experimental plan is well designed since a broad characterization was performed. The weakness of this work is the clear less in the exposition of the experimental design and materials and methods part, that doesn’t allow an immediate understanding of the study; furthermore conclusions are very poor. I suggest the attached revisions before publication,

Thanks for your appreciation. We have reviewed the manuscript point by point according to the reviewer.

  • Line 24-25: the sentence is not clear.

Thank you very much for your comment. The aforementioned phrase has been modified.

  • Line 47: insert prebiotic definition.

Thank you. The definition of prebiotic was inserted.

  • Line 82-88 paragraph “experimental design” is not clear and not fluent for reading. What is meant for enzymatic characteristics? Specify; bacterial resistance to what? specify; Specify the overall incubation time and the intermediate time for each parameters tested (as you did in the abstract) to clarify the experimental design. To follow a possible reorganization of the paragraph: “the probiotic viability, enzymatic characteristics (…..) and bacterial resistance to ………..( acid bile, lysozyme, gastric juices) of Lactobacillus acidophilus LYO 50( (Danisco, Dairy at 0, 30, 60, 90, 20 and 120 min of incubation…………………..……..All the experiments were done in triplicate with duplicate readings”

Thank you so much. The experimental design was modified.

  • Line 91: specify that bacterial viability was evaluated in presence and absence of Carao pulp at three different concentrations.

Thank you so much. Said appreciation was modified in the manuscript.

  • Line 97: specify that bile tolerance was evaluated in presence and absence of Carao pulp at three different concentrations.

Thank you so much. Said appreciation was modified in the manuscript.

  • Line 106: merge acid tolerance and resistance to gastric juice as you did in the results. specify acid tolerance and resistance to gastric juices was evaluated in presence and absence of Carao pulp at three different concentrations.

Thank you so much. Said appreciation was modified in the manuscript.

  • Line 111: specify protease activity was evaluated in presence and absence of Carao pulp at three different concentrations.

Thank you so much. Said appreciation was introduced in the text.

  • Line 114: …helping to maintain… what?

Thank you so much. That expression was deleted as it was an error

  • Line 130: specify that lysozyme tolerance was evaluated in presence and absence of Carao pulp at three different concentrations.

Thank you so much. This assessment was introduced.

  • Line 139: What is the difference between the paragraph: “bacterial viability” and “enumeration of L. acidophilus”? if you mean something different, specify, otherwise, you have to merge everything into only one paragraph. In any case the paragraph’s concept is not clear.

Thank you. Bacterial viability is inoculating the bacteria en MRS broth numeration of L. acidophilus is plating the bacteria with MRS agar. Paz et al., (2022) proposed the methods as well.

  • Line 224: specify also here that the results at different pH values are shown in the figure 4.

Thank you so much. Said modification was introduced in the aforementioned line.

  • Conclusion: expand the conclusion by reviewing the main results obtained for each tested parameter.

Thank you so much. The conclusions have been modified and expanded.

  • Line 304-305: in my opinion should be stressed the following Connection, Madison, WI) was studied in absence (Control) and in presence of three different concentration of Carao pulp powder (1%, 2%, and 3%). Acid and lysozyme tolerance was determined conclusion: even if Carao did not promote the microorganism growth (results obtained from viability test), its prebiotic effect is explicated through its protective effect for some of stressful conditions encountered during the digestive tract and for Carao effect in promoting proteolytic activity.

Thank you so much. Your comment was considered

Reviewer 2 Report

The manuscript entitled  “Attributes of Lactobacillus acidophilus as affected by Carao (Cassia grandis) pulp powder” is an interesting contribution to the study of Carao pulp powder and Lactobacillus acidophilus growth. The introduction is well written, the objective is clear, the material and methods are reproducible and the conclusions are supported by data, only minor details are needed previous to accepting the manuscript

Material and methods

The main concern is about the Carao powder, proximate analysis could be helping to discuss several findings

Results and discussions

Viability

The main concern about this topic is related to the effect, The viability increase is related to carbohydrates or related with fiber or related with bioactives or a mixture of all

L212 Again if there is no evaluation of the polysaccharides present in Carao powder is complicated discuss it

Author Response

Thanks for your appreciation. We have reviewed the manuscript point by point according to the reviewer.

Reviewer 2

Comments to Author

The manuscript entitled  “Attributes of Lactobacillus acidophilus as affected by Carao (Cassia grandis) pulp powder” is an interesting contribution to the study of Carao pulp powder and Lactobacillus acidophilus growth. The introduction is well written, the objective is clear, the material and methods are reproducible and the conclusions are supported by data, only minor details are needed previous to accepting the manuscript.

Thank you very much for your comments. The manuscript has been reviewed and their valuable contributions have been introduced.

Material and methods

The main concern is about the Carao powder, proximate analysis could be helping to discuss several findings

Results and discussions

Thank you so much. The chemical composition of the powder of the carao was not the objective of study in this work since in previous publications this characterization has been carried out. 10.5539/jas.v12n8p277

Viability

The main concern about this topic is related to the effect, The viability increase is related to carbohydrates or related with fiber or related with bioactives or a mixture of all

Thank you so much. The increase in viability is due to the synergistic effect of all.

L212 Again if there is no evaluation of the polysaccharides present in Carao powder is complicated discuss it

Thank you so much. Although the polysaccharides have not been identified, it is known from previous studies that this effect is due to total carbohydrates.

Round 2

Reviewer 1 Report

The revisions made are exhaustive. The article is publishable.